# SSA-Seg: Semantic and Spatial Adaptive Pixel-level Classifier for Semantic Segmentation

**Xiaowen Ma**[1,2*],    **Zhenliang Ni**[1*],    **Xinghao Chen**[1†]

[1]Huawei Noah's Ark Lab    [2]Zhejiang University

**https://github.com/xwmaxwma/SSA-Seg**

## Abstract

Vanilla pixel-level classifiers for semantic segmentation are based on a certain paradigm, involving the inner product of fixed prototypes obtained from the training set and pixel features in the test image. This approach, however, encounters significant limitations, *i.e.*, feature deviation in the semantic domain and information loss in the spatial domain. The former struggles with large intra-class variance among pixel features from different images, while the latter fails to utilize the structured information of semantic objects effectively. This leads to blurred mask boundaries as well as a deficiency of fine-grained recognition capability. In this paper, we propose a novel Semantic and Spatial Adaptive Classifier (SSA-Seg) to address the above challenges. Specifically, we employ the coarse masks obtained from the fixed prototypes as a guide to adjust the fixed prototype towards the center of the semantic and spatial domains in the test image. The adapted prototypes in semantic and spatial domains are then simultaneously considered to accomplish classification decisions. In addition, we propose an online multi-domain distillation learning strategy to improve the adaption process. Experimental results on three publicly available benchmarks show that the proposed SSA-Seg significantly improves the segmentation performance of the baseline models with only a minimal increase in computational cost.

## 1  Introduction

Semantic segmentation, as a fundamental task in computer vision, aims at assigning a category label to each pixel in a given image and is widely used in various domains such as autonomous driving [25], industrial detection [13], satellite image analysis [60, 27] and smart city [26]. Mainstream methods, such as FCN [34], DeepLab [5, 6], PSPNet [59], and SegNeXt [17] mainly use parametric softmax to classify each pixel. In recent developments, transformer-based approaches like MaskFormer [10], Mask2Former [9], and SegViT [55] classify masks by directly learning query vectors, termed as mask-level classification. However, these mask-level classification models often require heavyweight cross-attention decoders, which limit their deployment in resource-constrained scenarios [45, 57, 14, 35, 7].

Vanilla parametric softmax uses the convolutional kernel weights as the fixed semantic prototypes and obtains segmentation masks by computing the inner product of the pixel features and the prototypes. However, this pixel-level classification method has two obvious drawbacks: (1) *feature deviation in the semantic domain*. Due to complex backgrounds and varying object distributions, pixel features in the test images tend to have a large intra-class variance with pixel features in the training set. However, the fixed semantic prototypes, representing the semantic feature distribution on the training set, will be far away from the pixel features of the corresponding class when applied to the test image, as shown in Fig. 1(b). (2) *information loss in the spatial domain*. The vanilla pixel-level classifiers only perform

---

*Equal contribution,  † Corresponding author.

38th Conference on Neural Information Processing Systems (NeurIPS 2024).

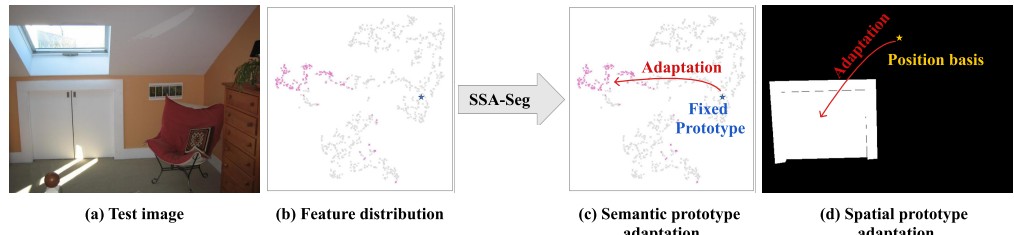

| (a) Test image | (b) Feature distribution | (c) Semantic prototype adaptation | (d) Spatial prototype adaptation |

Figure 1: A example of vanilla pixel-level classifiers, where the SeaFormer-L [45] is the baseline and the feature distribution is visualized with t-SNE. (a) is a test image of the ADE20K dataset, and (b) denotes the feature distributions in the semantic domain of (a), with purple and gray dots denoting the pixel features on the test image of the door and other categories, respectively. Blue star denotes the fixed prototype trained on training set of the door category. It shows that vanilla pixel-level classifiers directly interact pixel features with the fixed semantic prototypes, which leads to feature deviation in the semantic domain and information loss in the spatial domain problems. In contrast, SSA-Seg makes classification decisions based on adaptive semantic and spatial prototypes by prompting the prototypes to offset toward the center of the semantic domain and the spatial domain, as shown in (c) and (d). Visual comparison of the baseline and SSA-Seg can be found in Fig. 5.

inner products of fixed prototypes and pixel features in the semantic domain, and lack modeling of the relationship between prototypes and pixel features in the spatial domain. Therefore, the explicit structural information of target objects is not fully utilized, leading to suboptimal segmentation of border regions and small targets.

Recent studies [62, 28, 46] have improved the parametric softmax classifier. ProtoSeg [62] proposes a nonparametric prototype to replace the standard classifier. Through prototype-based metric learning, they enhanced the construction of pixel embedding space. GMMSeg [28] models the joint distribution of pixels and categories and learns the Gaussian mixture classifier in the pixel feature space by using the EM algorithm. However, these methods still rely on fixed prototype classifiers, when confronted with varying data distributions on test images, the problem of feature deviation in the semantic domain still exists. Besides, the Context-aware Classifier (CAC) [44] is proposed, which is a dynamic classifier by utilizes contextual cues. It adaptively adjusts classification prototypes based on feature content and thus can alleviate the feature deviation in the semantic domain problem to some extent. However, due to the lack of constraints on semantic prototypes, the offset of the fixed prototype is not controllable. In extreme cases, the offset will even be in the direction away from the semantic features of the image. Besides, these works ignore modeling the relationship between prototype and pixel features in the spatial domain, which limits further improvements in model performance.

To address the above issues, a Semantic and Spatial Adaptive Classifier (SSA-Seg) is proposed, which solves the above two problems by facilitating the offset of the fixed prototype towards the center of the semantic domain and the center of the spatial domain of the test image, as shown in Fig. 1(c) and (d). Note that the semantic domain center of each class is the mean semantic features belonging to this class, and similar concepts such as class center also appear in [56, 21]. Similarly, the center of the spatial domain is the mean spatial features belonging to the class. Specifically, we retain the original $1 \times 1$ convolution to obtain a coarse mask. Besides, position embedding is applied to the output features to obtain the spatial information. Then, the coarse mask is applied as the guide to obtain the semantic domain center and spatial domain center in each sample, which is then processed to obtain adaptive semantic and spatial prototypes. Finally, classification decisions are made by simultaneously considering prototypes in the semantic and spatial domains.

In addition, ground truth can be used to improve classifier performance [44]. Therefore, a training-only teacher classifier is designed to introduce ground truth information for calibrating the primary classifier. Specifically, we design multi-domain knowledge distillation to enhance the primary classifier from different domains. First, the response domain distillation method distills the outputs of the two classifiers based on a boundary-aware and category-aware distillation loss, which conveys accurate semantic and structural information in the ground truth. Then, semantic domain distillation and spatial domain distillation are used to constrain the offset of the prototype. In this way, multi-domain distillation improves the feature representation of the primary classifier, which significantly improves the test performance.

The proposed method significantly improves the segmentation performance of the baseline model with a very small increase in computational cost on three commonly used datasets: ADE20k, PASCAL-Context, and COCO-stuff-10K. Furthermore, compared to other advanced classifiers

such as GMMSeg [28] and CAC [44], our SSA-Seg achieves significant performance improvement on several baseline methods. In particular, by applying SSA-Seg, we achieve the state-of-the-art lightweight segmentation performance. Specially, SeaFormer-L [45] obtain 45.36% mIoU on ADE20k with only 0.1G more FLOPs and 0.5ms more latency. Simarlily, SegNext-T [17] obtain 38.91% mIoU on COCO-stuff-10K and 52.58% mIoU on PASCAL-Context dataset.

## 2 Related Work

### 2.1 Semantic Segmentation

Semantic segmentation is one of the basic tasks in computer vision, whose goal is to assign a category to each pixel in an image. The existing semantic segmentation methods can be divided into two categories: pixel-level classification model [34, 54, 45, 57, 14, 17] and mask-level classification model [10, 9, 55, 18, 31, 51]. Since Fully Convolutional Networks (FCNs) [34], pixel-level classification has been the mainstream semantic segmentation method. Subsequent works focus on optimizing backbone extraction features [17, 45, 8, 52, 33, 53, 50], or improving the decode head for context modeling [37, 36, 59, 6, 15, 22, 54, 24, 29].

Recently, mask-level classification models have become popular, which learn object queries to classify masks without classifying each pixel. MaskFormer [10] first proposes to use mask classification for semantic segmentation tasks, which is inspired by [3]. Mask2Former [9] optimizes MaskFormer [10] by constraining cross-attention within the predicted masked region to extract local features. SegViT [55] proposes to apply the mask classifier to plain vision transformers. However, mask-level classification models often have high computational complexity due to the need for multiple cross-attention to update the query, and their application fields are limited. Pixel-level classification model, as the mainstream method, has a wide range of application fields. Therefore, we rethink pixel-level classifiers to improve the performance of pixel-level classification.

### 2.2 Pixel-level Classifier

The current mainstream Pixel-level classifier is essentially a discriminant classification model based on softmax. The potential data distribution is completely ignored, which limits the expressive ability of the model. Recently, some works [62, 28, 44] have improved the softmax classifier to improve the segmentation performance. ProtoSeg [62] proposes a non-parametric prototype to replace the standard classifier. Through prototype-based metric learning, the construction of pixel embedding space is better. GMMSeg [28] models the joint distribution of pixels and categories and learns the Gaussian mixture classifier in the pixel feature space by using the EM algorithm. It can capture the pixel feature distribution of each category in fine detail by using the generator pattern. Context-aware classifer [44] extracts the context based on the input to generate a sample adaptive classifier. However, none of these methods use position embedding to improve the classifier. Our classifier can sense both semantic and spatial information, to make more accurate classification decisions.

## 3 Method

### 3.1 Preliminary: Vanilla Pixel-level Classifier

We first review the architecture of the vanilla pixel-level classifier for semantic segmentation. For an input image $\mathcal{X}$, the features are first extracted by backbone and processed by decode head to obtain the output features $\mathcal{S}_f \in \mathbb{R}^{H \times W \times D}$, where $H$, $W$, $D$ denote the height, width, and channel of the feature map, respectively. The current mainstream strategy uses a fixed prototype classifier, *i.e.*, a simple $1 \times 1$ convolution maps feature to the category space:

$$\mathcal{M}_c = \mathcal{S}_f \otimes \mathcal{S}^T, \tag{1}$$

where $\mathcal{M}_c \in \mathbb{R}^{H \times W \times K}$ denotes the segmentation mask, $\mathcal{S} \in \mathbb{R}^{K \times D}$ denotes the weights of $1 \times 1$ convolution kernel, $\otimes$ denotes matrix multiplication and $K$ is the number of category.

The above scheme essentially treats the convolutional kernel weights as fixed semantic prototypes and obtains segmentation masks based on the inner product with pixel features in the semantic domain. Therefore, the two questions of *feature deviation in the semantic domain* and *information loss in the spatial domain*, which have been described before, are to be solved.

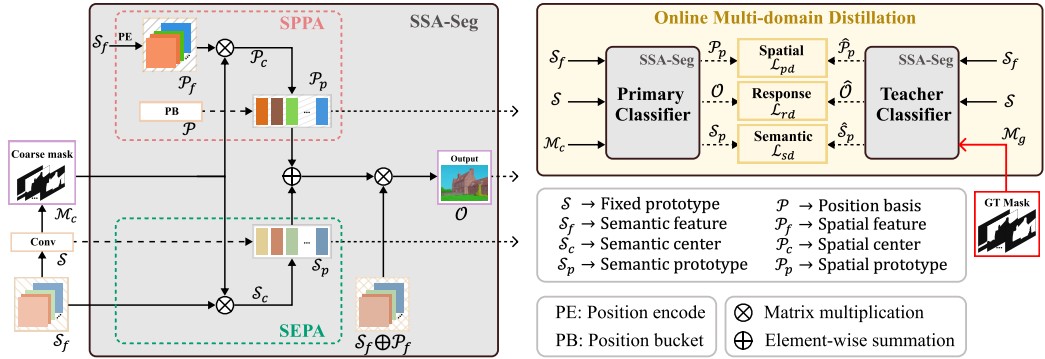

Figure 2: SSA-Seg overview. For the semantic features $\mathcal{S}_f$ output from the backbone and decode head, we first generate spatial features $\mathcal{P}_f$ by position encode. Then we retain the original $1 \times 1$ convolution to generate the coarse mask $\mathcal{M}_c$. Guided by $\mathcal{M}_c$, we generate the center of the semantic domain and spatial domain in the pre-classified representations and fused them with the fixed semantic prototypes $\mathcal{S}$ and the prototype position basis $\mathcal{P}$ to generate the semantic prototypes $\mathcal{S}_p$ and the spatial prototype $\mathcal{P}_p$. Finally, we consider simultaneously semantic and spatial prototypes to perform classification decisions. The right figure shows an online teacher classifier only for training, where the coarse mask is replaced with ground-truth mask to participate in model training, and constrains the prototype adaption and transfer accurate semantic and spatial knowledge to the primary classifier based on multi-domain distillation learning.

## 3.2 Semantic and Spatial Adaptive Classifier

Based on the above analysis in Section 3.1, we propose a novel Semantic and Spatial Adaptive Classifier (SSA-Seg), as shown in Fig. 2. The proposed SSA-Seg consists of three parts: semantic prototype adaptation, spatial prototype adaptation, and online multi-domain distillation.

Specifically, for the semantic features $\mathcal{S}_f$ output from the backbone and the decode head, we retain the original $1 \times 1$ convolution operation to generate the coarse mask $\mathcal{M}_c$. Next, we obtain the spatial features $\mathcal{P}_f$ from the semantic features $\mathcal{S}_f$. Guided by $\mathcal{M}_c$, we integrate the semantic and spatial features of each category to obtain the centers of the corresponding semantic and spatial domains, i.e., $\mathcal{S}_c$ and $\mathcal{P}_c$. Then, we fuse $\mathcal{S}_c$, $\mathcal{P}_c$ with the fixed semantic prototypes $\mathcal{S}$ and prototype position basis $\mathcal{P}$, respectively, to generate adaptive semantic prototypes $\mathcal{S}_p$ and spatial prototypes $\mathcal{P}_p$. Finally, we modify Eq. 1 to make more accurate classification decisions,

$$\mathcal{O}_c = (\mathcal{S}_f \oplus \mathcal{P}_f) \otimes (\mathcal{S}_p \oplus \mathcal{P}_p)^T, \tag{2}$$

where $\oplus$ denotes element-wise summation. In addition, we introduce a teacher classifier and an online multi-domain distillation strategy to improve the performance of SSA-Seg. The teacher classifier has the same structure as the primary classifier, except that the coarse mask $\mathcal{M}_c$ is replaced by the ground-truth mask $\mathcal{M}_g$, as shown in Fig. 2. Therefore, the teacher classifier can output more accurate semantic prototypes $\hat{\mathcal{S}}_p$, spatial prototypes $\hat{\mathcal{P}}_p$ and segmentation masks $\hat{\mathcal{O}}$. These outputs can be used as soft labels to constrain the adaptation process of prototypes through online multi-domain distillation, which consists of response domain distillation loss $\mathcal{L}_{rd}$, semantic domain distillation loss $\mathcal{L}_{sd}$, and spatial domain distillation loss $\mathcal{L}_{pd}$. Therefore, the training loss $\mathcal{L}$ is,

$$\mathcal{L} = \mathcal{L}_c + \mathcal{L}_g + \lambda_r \mathcal{L}_{rd} + \lambda_s \mathcal{L}_{sd} + \lambda_p \mathcal{L}_{pd}, \tag{3}$$

where $\mathcal{L}_c = \mathcal{L}_{ce}^c + \mathcal{L}_{dice}^c$, $\mathcal{L}_g = \mathcal{L}_{ce}^g + \mathcal{L}_{dice}^g$, $\mathcal{L}_{ce}^c$ and $\mathcal{L}_{ce}^g$ denote cross-entropy loss for $\mathcal{O}$ and $\hat{\mathcal{O}}$, respectively. Similarly, $\mathcal{L}_{dice}^c$ and $\mathcal{L}_{dice}^g$ denote dice loss [38] for $\mathcal{O}$ and $\hat{\mathcal{O}}$, respectively.

### 3.2.1 Semantic Prototype Adaptation.

Previous work [44] utilizes additional contextual cues to offset the fixed prototypes and thus adapt to the semantic feature distributions of different images, alleviating the feature deviation in the semantic domain problem to some extent. However, due to the lack of constraints, the offset of the fixed prototype is not controllable. In extreme cases, the offset will even be in the direction away from the semantic features (An example visualization can be found in Fig. 6.). Therefore, we propose SEmantic Prototype Adaptation (SEPA), which offsets fixed semantic prototypes based on coarse mask-guided semantic feature distributions, and constrains the adaptive semantic prototype $\mathcal{S}_p$ with semantic domain distillation.

Specifically, we first multiply the semantic features $\mathcal{S}_f$ and the coarse mask $\mathcal{M}_c$ to obtain the semantic domain center $\mathcal{S}_c$, *i.e.*, $\mathcal{S}_c = \text{Softmax}_K(\mathcal{M}_c) \otimes \mathcal{S}_f$, where $\mathcal{S}_c \in \mathbb{R}^{K \times D}$ denotes the average features of all pixel features belonging to different classes of the coarse mask-guided pre-classified representations. We then concatenate $\mathcal{S}_c$ with the fixed prototype $\mathcal{S}$, and fuse them through a $1 \times 1$ convolution layer $\phi_s$ to obtain the adaptive semantic prototype $\mathcal{S}_p \in \mathbb{R}^{K \times D}$,

$$\mathcal{S}_p = \phi_s(\mathcal{S}_c \odot \mathcal{S}), \tag{4}$$

where $\odot$ denotes channel concatenation operation. With the above operation and the constraint of semantic domain distillation loss, the fixed prototype offsets towards the semantic feature distribution of the test image. As a result, the pixel features possess higher similarity to the semantic prototypes of the corresponding categories, which promotes more pixels to be correctly categorized.

### 3.2.2 Spatial Prototype Adaptation.

Previous classifiers [34, 28, 62] are mainly based on the inner product of pixel features and prototypes in the semantic domain, without utilizing the rich spatial information of the image. However, most of the target objects in semantic segmentation tasks possess regular shapes, such as doors, windows, and roads. Modeling the spatial relations of pixel and prototype can introduce structured information about the target objects, thus improving the segmentation performance for boundary regions and small targets. Therefore, we introduce spatial prototype adaptation (SPPA), which aims to make classification decisions with additional consideration of the spatial relation between pixel features and prototypes.

We first obtain the spatial features $\mathcal{P}_f \in \mathbb{R}^{H \times W \times D}$ with position encoding of the pixel features $\mathcal{S}_f$, *i.e.*, $\mathcal{P}_f = \text{PE}(\mathcal{S}_f)$. Here we choose conditional position encoding [11], which is useful for encoding neighborhood information to further localize the mask region. Similarly, we perform matrix multiplication based on the coarse mask with the feature position coding to obtain $\mathcal{P}_c$,

$$\mathcal{P}_c = \text{Softmax}_{HW}(\mathcal{M}_c) \otimes \mathcal{P}_f, \tag{5}$$

where $\mathcal{P}_c \in \mathbb{R}^{K \times D}$ denotes the spatial domain center of the coarse mask-guided pre-classified representations. Note that here we implement the softmax function for $\mathcal{M}_c$ in spatial dimensions, which facilitates modeling the spatial distribution of the different categories and thus the spatial location of the category prototypes on the image.

In addition, only a few categories appear on the image in most cases. Therefore, only the positions of the corresponding categories in the $\mathcal{P}_c$ have practical significance. In order to maintain the stability of training, we define a randomly initialized position basis $\mathcal{P} \in \mathbb{R}^{K \times D}$ and concatenate it with $\mathcal{P}_c$ to obtain the spatial prototype $\mathcal{P}_p \in \mathbb{R}^{K \times D}$, after mapping with a $1 \times 1$ convolution layer $\phi_p$,

$$\mathcal{P}_p = \phi_p(\mathcal{P}_c \odot \mathcal{P}). \tag{6}$$

Similarly, based on the above operation and the constraint of spatial domain distillation loss, the prototype position basis is offset towards the center of the mask region for each class of the test image, thus generating the spatial prototype $\mathcal{P}_p$. Therefore, the spatial relationship of the pixel features with the prototype can be taken into account when making the classification decision, which improves the segmentation performance of the boundaries and the small target regions. This is verified by the qualitative and quantitative analysis in Section 4.2.

### 3.2.3 Online Multi-Domain Distillation Learning.

Although semantic and spatial domain adaptations can motivate better interaction of prototypes with pixel features, the offset of semantic and spatial prototypes is not controllable due to the lack of constraint, which affects the segmentation performance of the model. In this paper, we propose an online multi-domain distillation learning to optimize the process of feature generation and constrain the adaptation of the semantic and spatial prototype.

Specifically, different from the previous widely adopted offline distillation learning method [30, 42, 47, 58, 23], we incorporate ground truth directly into the model training process [44] and construct soft labels, which can convey useful information to the model and do not require the additional process of training teacher models. As shown in Fig. 2, we first create a new branch with pixel features $\mathcal{S}_f$ as inputs and use the ground truth $\mathcal{M}_g$ to guide the adaption process of semantic and spatial prototypes, which generates the semantic prototype $\hat{\mathcal{S}}_p$, spatial prototype $\hat{\mathcal{P}}_p$, and segmentation mask $\hat{\mathcal{O}}$. $\hat{\mathcal{P}}_p$, $\hat{\mathcal{S}}_p$, and $\hat{\mathcal{O}}$ are used as the online teachers to distillate $\mathcal{P}_p$, $\mathcal{S}_p$, and $\mathcal{O}$, respectively.

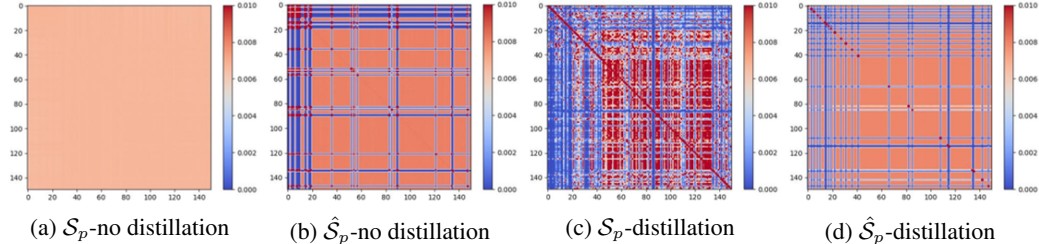

(a) $\mathcal{S}_p$-no distillation     (b) $\hat{\mathcal{S}}_p$-no distillation     (c) $\mathcal{S}_p$-distillation     (d) $\hat{\mathcal{S}}_p$-distillation

Figure 3: Visualization of the inter-class relation matrix for the semantic prototypes $\mathcal{S}_p$ and $\hat{\mathcal{S}}_p$, and the latter possesses better inter-class separability. This motivates us to add semantic domain distillation loss to constrain the adaption of the semantic prototypes. The results show that after semantic domain distillation, the semantic prototypes have better separability, which facilitates category recognition.

**Response Domain Distillation.** The ground truth guided segmentation mask $\hat{\mathcal{O}}$ has a higher entropy value compared to the one-hot label, which can provide more information to the model. We first give the expression for the original response domain distillation learning,

$$\mathcal{L}_{rd}^i = -\sum_{j=1}^{K} \psi(\hat{\mathcal{O}})^{i,j} \cdot log(\psi(\mathcal{O}^{i,j})), \quad \mathcal{L}_{rd} = \frac{-1}{HW} \sum_{i=1}^{HW} \mathcal{L}_{rd}^i, \tag{7}$$

where $\psi$ denotes the Softmax function along the channel dimension. However, the original expression averages the distillation loss for pixels at all spatial locations, which results in the spatial structural information of the object and the semantic information of a few sample categories being masked by pixels at other locations. Therefore, we design a boundary-aware and category-aware distillation loss to induce the transfer of semantic and spatial information from $\hat{\mathcal{O}}$ to the prediction masks $\mathcal{O}$.

Specifically, we first obtain the semantic mask $\mathcal{E}$ via $\mathcal{M}_g$. We then use the Canny operation to extract the boundary of $\mathcal{E}$ and obtain the boundary mask $\mathcal{B}$. Based on $\mathcal{E}$ and $\mathcal{B}$, we modify Eq. 7,

$$\mathcal{L}_{rd} = \frac{-1}{2K} \sum_{k=1}^{K} \left( \frac{\sum_{i=1}^{HW} \mathcal{L}_{rd}^i \cdot \mathcal{B}_k^i}{\sum_{i=1}^{HW} \mathcal{B}_k^i} + \frac{\sum_{i=1}^{HW} \mathcal{L}_{rd}^i \cdot \bar{\mathcal{B}}_k^i}{\sum_{i=1}^{HW} \bar{\mathcal{B}}_k^i} \right), \tag{8}$$

where $\mathcal{B}_k = \mathcal{E}_k \cdot \mathcal{B}$ denotes the boundary mask of class $k$, and $\bar{\mathcal{B}}_k$ denotes the non-boundary mask of class $k$. Finally, we retain the entropy aware of [44] in order to adjust the contribution of each element according to the level of information,

$$\mathcal{L}_{rd} = \frac{-1}{2K} \sum_{k=1}^{K} \left( \frac{\sum_{i=1}^{HW} \mathcal{L}_{rd}^i \mathcal{B}_k^i \mathcal{H}^i}{\sum_{i=1}^{HW} \mathcal{B}_k^i \mathcal{H}^i} + \frac{\sum_{i=1}^{HW} \mathcal{L}_{rd}^i \cdot \bar{\mathcal{B}}_k^i \mathcal{H}^i}{\sum_{i=1}^{HW} \bar{\mathcal{B}}_k^i \mathcal{H}^i} \right), \quad \mathcal{H}^i = -\sum_{j=1}^{K} \psi(\hat{\mathcal{O}})^{i,j} \cdot log(\psi(\hat{\mathcal{O}}^{i,j})) \tag{9}$$

where $\mathcal{H}^i$ denotes the entropy of the i-th pixel prediction of $\hat{\mathcal{O}}$.

**Semantic Domain Distillation.** We propose the semantic domain distillation loss $\mathcal{L}_{sd}$ to guide the offset process of semantic prototypes. As shown in Fig. 3 (a) and (b), the ground truth guided semantic prototype $\hat{\mathcal{S}}_p$ exhibits better inter-class separation properties compared to the $\mathcal{S}_p$. In fact, the inter-class separation properties are critical for optimizing the feature embedding space and making classification decisions. Previous work [58] focuses on constraining inter-class relationships to be identical, which is difficult to train and leads to poor generalization. Intuitively, for inter-class relationships, we only need to constrain the closer classes in the student model to be farther away. For those classes that are further away compared to the teacher model, we do not need to negatively supervise them. Specifically, we first compute the inter-class similarity matrix of semantic prototypes,

$$\mathcal{M} = \psi(\mathcal{S}_p \mathcal{S}_p^T), \quad \hat{\mathcal{M}} = \psi(\hat{\mathcal{S}}_p \hat{\mathcal{S}}_p^T). \tag{10}$$

Then, the difference between $\mathcal{M} \in \mathbb{R}^{K \times K}$ and $\hat{\mathcal{M}} \in \mathbb{R}^{K \times K}$ is calculated as,

$$\mathcal{M}_d = \Lambda(\mathcal{M} - \hat{\mathcal{M}}), \tag{11}$$

where $\Lambda$ is a mask operation that sets the value of both the diagonal position and the position less than zero to zero. Finally, we compute the semantic domain distillation loss $\mathcal{L}_{sd}$,

$$\mathcal{L}_{sd} = \frac{1}{K} \sum_{i=1}^{K} \sum_{j=1}^{K} \mathcal{M}_d^{i,j}. \tag{12}$$

Table 1: Performance comparison of SSA-Seg on state-of-the-art general (top) and light weight (bottom) methods. The number of FLOPs (G) is calculated on the input size of $512 \times 512$ for ADE20K and COCO-Stuff-10K, and $480 \times 480$ for PASCAL-Context. The latency (ms) is calculated on the input size of $512 \times 512$ on V100 GPU. The green number indicates the increase from the baseline.

| Method | Backbone | Latency | Params | ADE20K | | COCO-Stuff-10K | | PASCAL-Context | |
|---|---|---|---|---|---|---|---|---|---|
| | | | | FLOPs | mIoU | FLOPs | mIoU | FLOPs | mIoU |
| OCRNet [54] | HRNet-W48 | 67.2 | 8.6 | 164.8 | 43.30 | 164.8 | 36.16 | 143.2 | 48.22 |
| +SSA-Seg | | 69.3 | 8.7 | 165.0 | 47.47 ↑4.17 | 165.0 | 37.94 ↑1.78 | 143.3 | 50.21 ↑1.99 |
| UperNet [48] | Swin-T | 52.8 | 60.0 | 236.1 | 44.14 | 236.1 | 38.93 | 207.5 | 51.93 |
| +SSA-Seg | | 54.3 | 61.1 | 236.3 | 47.56 ↑3.42 | 236.3 | 42.30 ↑3.37 | 207.7 | 54.91 ↑2.98 |
| SegFormer [49] | MiT-B5 | 69.0 | 82.0 | 52.5 | 49.13 | 52.5 | 44.07 | 45.8 | 58.39 |
| +SSA-Seg | | 70.1 | 82.3 | 52.6 | 50.74 ↑1.61 | 52.6 | 45.55 ↑1.48 | 45.8 | 59.14 ↑0.75 |
| UperNet [48] | Swin-L | 105.5 | 233.8 | 404.9 | 51.68 | 404.9 | 46.85 | 362.9 | 60.50 |
| +SSA-Seg | | 107.3 | 234.9 | 405.2 | 52.69 ↑1.01 | 405.2 | 48.94 ↑2.09 | 363.2 | 61.83 ↑1.33 |
| ViT-Adapter [8] | ViT-Adapter-L | 283.3 | 363.8 | 616.1 | 54.40 | 616.1 | 50.16 | 541.5 | 65.77 |
| +SSA-Seg | | 284.9 | 364.9 | 616.3 | 55.39 ↑0.99 | 616.3 | 51.18 ↑1.02 | 541.7 | 66.05 ↑0.28 |
| AFFormer-B [14] | AFFormer-B | 25.1 | 3.0 | 4.3 | 39.94 | 4.3 | 33.22 | 3.7 | 48.57 |
| +SSA-Seg | | 26.0 | 3.3 | 4.4 | 41.92 ↑1.98 | 4.4 | 36.40 ↑3.18 | 3.7 | 49.72 ↑1.15 |
| SeaFormer-B [45] | SeaFormer-B | 26.8 | 8.6 | 1.8 | 40.05 | 1.8 | 33.29 | 1.6 | 45.75 |
| +SSA-Seg | | 27.3 | 8.8 | 1.8 | 42.46 ↑2.41 | 1.8 | 35.92 ↑2.63 | 1.6 | 47.00 ↑1.25 |
| SegNeXt-T [17] | MSCAN-T | 22.8 | 4.3 | 6.2 | 41.04 | 6.2 | 36.39 | 5.4 | 50.35 |
| +SSA-Seg | | 23.3 | 4.6 | 6.3 | 43.90 ↑2.86 | 6.3 | 38.91 ↑2.52 | 5.4 | 52.58 ↑2.23 |
| SeaFormer-L [45] | SeaFormer-L | 29.4 | 14.0 | 6.4 | 42.36 | 6.4 | 35.99 | 5.6 | 49.14 |
| +SSA-Seg | | 29.9 | 14.2 | 6.4 | 45.36 ↑3.00 | 6.4 | 38.48 ↑2.44 | 5.6 | 49.66 ↑0.52 |

Note that $\hat{S}_p$ in Eq. 10 is obtained based on the average semantic domain center of all training images in a batch. It can be used to minimize the negative impact of noisy images and accelerate model convergence.

With semantic domain distillation, the semantic prototype $\mathcal{S}_p$ exhibits similar separability to $\hat{\mathcal{S}}_p$, which facilitates the recognition of categories, as shown in Fig. 3 (c) and (d).

**Spatial Domain Distillation.** Unlike semantic domain distillation, since spatial domain distillation focuses on the spatial structure information of an object, we tend to constrain the spatial prototypes guided by the rough mask to be equal to the spatial prototypes guided by the ground-truth mask,

$$\mathcal{L}_{pd} = \frac{-1}{K} \sum_{i=1}^{K} \sum_{j=1}^{D} \psi(\mathcal{P}_p)^{i,j} \cdot log(\psi(\hat{\mathcal{P}}_p^{i,j})).$$ (13)

# 4 Experiments

We perform experiments on the ADE20K [61], PASCAL-Context [39] and COCO-Stuff-10K [1] datasets. We use MMSegmentation [12] and follow the common training settings. Please refer to the Appendix for more details.

## 4.1 Main Results

**Efficiency Comparison.** We first focus on the efficiency changes introduced by SSA-Seg, as shown in Table 1. It can be observed that the additional FLOPs and latency introduced by SSA-Seg is negligible for methods of different sizes. For example, for general semantic segmentation methods such as ViT-Adapter-L, we only increase latency by 0.56% and flops by 0.03%, while for lightweight methods such as SeaFormer-L, we only increase latency by 1.70% and flops by 0.78%. It can be attributed to the fact that we add only a depth-wise convolution and some $1 \times 1$ convolutions to the primary classifier without changing the backbone and segmentation head. Therefore, the increased memory consumption, FLOPs and latency of SSA-Seg are negligible compared to the original model.

**Performance on state-of-the-art general methods.** As shown in Table 1, the proposed SSA-Seg can significantly improve the segmentation performance of various general models with negligible decrease in efficiency. For example, the application of SSA-Seg on UperNet-Swin-Tiny and ViT-Adapter-L bring 3.42% and 1.01% mIoU performance improvements on ADE20K dataset, respectively.

**Performance on state-of-the-art lightweight methods.** SSA-Seg helps methods such as SeaFormer and SegNeXt to achieve more superior performance without compromising efficiency. As shown in Table 1, by applying SSA-Seg, SeaFormer-L achieves 45.36% mIoU on ADE20K, while SegNext-T achieves 38.91% and 52.58% mIoU on COCO-Stuff-10K and PASCAL-Context datasets, respectively. This is the state-of-the-art performance obtained in real-time segmentation tasks.

**Comparison with state-of-the-art classifiers.** To further prove its superior performance, the SSA-Seg is compared with other state-of-the-art classifiers, as shown in Table 2. When UperNet is used as the baseline, the proposed SSA-Seg can improve the mIoU by 1.20%, while GMMSeg and DNC only increase mIoU by 1.00% and 0.60%, respectively. When OCRNet is used as the baseline, SSA-Seg exceeds baseline by 4.2% mIoU, which far exceeds GMMSeg by 2.7% mIoU. Also, SSA-Seg exceeds CAC 1.8% mIoU. For

Table 2: Comparison with other state-of-the-art classifiers. FLOPs (G) are calculated using the input size of $512 \times 512$ on the ADE20K dataset.

| Method | Backbone | FLOPs | ADE20K | COCO. |
|---|---|---|---|---|
| FCN [34] | | 275.7 | 39.9 | 32.5 |
| +ProtoSeg [62] | ResNet101 [19] | 278.5 | 41.1 $\uparrow_{1.2}$ | 34.0 $\uparrow_{1.5}$ |
| +DNC [46] | | 278.5 | 41.1 $\uparrow_{1.2}$ | 33.0 $\uparrow_{0.5}$ |
| **SSA-Seg** | | 275.9 | **44.3** $\uparrow_{4.4}$ | **36.6** $\uparrow_{4.1}$ |
| UperNet | | 297.2 | 48.0 | 42.8 |
| +GMMSeg [28] | Swin-B [32] | 302.3 | 49.0 $\uparrow_{1.0}$ | 44.3 $\uparrow_{1.5}$ |
| +DNC [46] | | 308.6 | 48.6 $\uparrow_{0.6}$ | 43.1 $\uparrow_{0.3}$ |
| **+SSA-Seg** | | 297.5 | **49.2** $\uparrow_{1.2}$ | **45.2** $\uparrow_{2.4}$ |
| OCRNet [54] | | 164.8 | 43.3 | 36.2 |
| +GMMSeg [28] | HRNetV2-W48 [43] | 169.8 | 44.8 $\uparrow_{1.5}$ | - |
| +CAC [44] | | 164.9 | 45.7 $\uparrow_{2.4}$ | - |
| **+SSA-Seg** | | 165.0 | **47.5** $\uparrow_{4.2}$ | **37.9** $\uparrow_{1.7}$ |
| SegNeXt-T [17] | | 6.2 | 41.0 | 36.4 |
| +CAC [44] | MSCAN-T [17] | 6.2 | 43.0 $\uparrow_{2.0}$ | 37.5 $\uparrow_{1.1}$ |
| **+SSA-Seg** | | 6.3 | **43.9** $\uparrow_{2.9}$ | **38.9** $\uparrow_{2.5}$ |
| SeaFormer-B [45] | | 1.8 | 40.0 | 33.3 |
| +CAC [44] | SeaFormer-B [45] | 1.8 | 40.1 $\uparrow_{0.1}$ | 35.5 $\uparrow_{2.2}$ |
| **+SSA-Seg** | | 1.8 | **42.5** $\uparrow_{2.5}$ | **35.9** $\uparrow_{2.6}$ |

SegNeXt-T, SSA-Seg exceeds CAC by 0.9% mIoU. Moreover, for the lightweight model SeaFormer, CAC cannot bring growth. SSA-Seg can still achieve 2.5% mIoU growth. The above experimental results demonstrate that the proposed SSA-Seg achieves state-of-the-art performance.

**Comparison with state-of-the-art mask classification models.** In order to validate the effectiveness of SSA-Seg, we further perform a comprehensive comparison with the state-of-the-art mask classification methods, as shown in Table 3. It should be noted that the DPG Head in CGRSeg [40] conflicts with SSA-Seg, and we remove the DPG Head before adding SSA-Seg.Therefore, compared with CGRSeg, CGRSeg+SSA-Seg rather reduces the efficiency metrics such as parameters, FLOPs and latency. The results show that by combining SSA-Seg, the existing pixel-level segmentation baselines achieve a better balance between efficiency and performance compared to mask classification methods.

Table 3: Comparison with state-of-the-art mask classification models. FLOPs (G) are calculated using the input size of $512 \times 512$. The latency (ms) is measured on a single V100 GPU with input size 512x512.

| Method | Params | FLOPs | Latency | mIoU |
|---|---|---|---|---|
| MaskFormer [10] | 41.3 | 55.1 | 31.0 | 44.5 |
| Mask2Former [9] | 44.0 | 70.1 | 55.2 | 47.2 |
| YOSO [20] | 42.0 | 37.3 | 28.3 | 44.7 |
| PEM [4] | 35.6 | 46.9 | 26.8 | 45.5 |
| CGRSeg-B [40] | 19.1 | 7.7 | 36.0 | 45.5 |
| **+SSA-Seg** | 19.3 | 7.6 | 36.0 | 47.1 |
| CGRSeg-L [40] | 35.7 | 14.9 | 43.3 | 48.3 |
| **+SSA-Seg** | 35.8 | 14.8 | 42.6 | 49.0 |

## 4.2 Ablation Study

**Ablation of the position basis.** We first explore the necessity of spatial prototype adaptation. As shown in Table 4, *Baseline+basis only* means only randomly initialized position basis is used and the baseline is SeaFormer-L. It can be observed that when only the positional basis is retained, the performance of the model is degraded due to the lack of a spatial prototype adaptation process and the corresponding spatial domain distillation.

Table 4: Ablation of the position basis.

| Method | Params | FLOPs | Latency | mIoU |
|---|---|---|---|---|
| SSA-Seg | 14.2 | 6.4 | 29.9 | 45.4 |
| Baseline+basis only | 14.2 | 6.4 | 29.8 | 44.3 |

**Ablation of Spatial domain center.** We also conduct an ablation experiment on the spatial domain center, as shown in Table 6. The results show that applying softmax in the spatial dimension gives better performance. This can be interpreted as the model learns the relative spatial distribution of each category and thus models a more accurate spatial domain center.

**Ablation of PE.** We explore the effect of different position encoding methods on the ADE20K dataset in Table 7. We chose three widely used methods, namely Sinusoidal position coding [3], learnable absolute position coding [16], and feature-dependent conditional position coding, *i.e.*, CPVT [11].

Table 5: Alation experiments of the response domain distillation loss $\mathcal{L}_{rd}$.

| Method | mIoU |
|--------|------|
| SSA-Seg | 45.36 |
| -$\mathcal{H}$ | 44.82 |
| -$\mathcal{B}$ | 44.77 |
| -$\mathcal{E}$ | 45.11 |

Table 6: Ablation experiments of the generation of $\mathcal{P}_c$. Softmax$_{HW}$ denotes apply softmax operation on spatial dimension.

| Method | mIoU |
|--------|------|
| Softmax$_K$ | 44.86 |
| Softmax$_{HW}$ | 45.36 |

Table 7: Ablation experiments of position encoding (PE) methods on the ADE20K dataset.

| PE | mIoU |
|----|------|
| Sinusoidal [3] | 44.18 |
| Learnable [16] | 44.02 |
| CPVT | 45.36 |

The results show that CPVT is particularly effective in improving the results. This can be explained by the fact that CPVT is able to encode neighborhood information and preserve the implicit positional priors to locate the region of interest at the center of the spatial domain for the semantic mask. In contrast, sinusoidal and learnable absolute position encoding can only describe one anchor point, which is not conducive to the localization of segmented fragments.

**Ablation of structure.** A series of ablation experiments are performed to verify the validity of the SEmantic Prototype Adaption (SEPA) and the SPatial Prototype Adaption (SPPA), respectively. The experimental results are shown in Table 8. The 1×1 convolution with softmax is used as the baseline. It can be found that the baseline achieves only 42.36% mIoU. By applying SEPA, mIoU increases by 0.52% with a growth of less than 0.1 GFlops. When combined with $\mathcal{L}_{sd}$ and $\mathcal{L}_g$, mIoU increases by 1.90%. This validates the necessity of semantic domain distillation to constrain the semantic prototype adaptation process, *i.e.*, they are structurally inseparable. Similarly, By applying SPPA and $\mathcal{L}_g + \mathcal{L}_{pd}$, mIoU increases by 2.12% and Flops increases by 0.1 G. The experimental results show that SEPA and SPPA can improve the segmentation performance.

Table 8: Ablation experiment of SEPA and SPPA.

| Method | FLOPs | mIoU |
|--------|-------|------|
| Baseline | 6.4 | 42.36 |
| + SEPA | 6.4 | 42.88 |
| + SPPA | 6.4 | 42.76 |
| + SEPA + SPPA | 6.4 | 43.27 |
| + SEPA + $\mathcal{L}_g$ + $\mathcal{L}_{sd}$ | 6.4 | 44.26 |
| + SPPA + $\mathcal{L}_g$ + $\mathcal{L}_{pd}$ | 6.4 | 44.48 |
| **SSA-Seg** | 6.4 | **45.36** |

**Effect of multi-domain knowledge distillation.** Multi-domain knowledge distillation can significantly improve the feature representation of SSA-Seg. To verify its performance, we conduct related ablation experiments. The experimental results are shown in Table 9. In the baseline, only the primary classifier is used, *i.e.*, without teacher classifier. $\mathcal{L}_g$ can increase mIoU by 0.60%. By applying response domain distillation $\mathcal{L}_{rd}$, semantic domain distillation $\mathcal{L}_{sd}$, and spatial domain distillation $\mathcal{L}_{pd}$, the mIoU can be further increased by 0.67%, 1.09% and 0.84%, respectively. Multi-domain distillation as a whole can achieve a 2.09% mIoU improvement. The above experimental results show that each domain distillation can improve the accuracy, and multi-domain distillation can further significantly improve the model performance. Furthermore, we perform an ablation analysis of $\mathcal{L}_{rd}$ design, as shown in Table 5. The results show that the introduction of entropy $\mathcal{H}$, category mask $\mathcal{E}$, and boundary mask $\mathcal{B}$ has a role to play in the improvement of the model performance.

Table 9: Ablation experiment of online multi-domain knowledge distillation.

| Method | mIoU |
|--------|------|
| Baseline + SEPA + SPPA | 43.27 |
| + $\mathcal{L}_g$ | 43.87 |
| + $\mathcal{L}_g$ + $\mathcal{L}_{rd}$ | 44.54 |
| + $\mathcal{L}_g$ + $\mathcal{L}_{sd}$ | 44.96 |
| + $\mathcal{L}_g$ + $\mathcal{L}_{pd}$ | 44.71 |
| + $\mathcal{L}_g$ + $\mathcal{L}_{sd}$+ $\mathcal{L}_{pd}$ | 45.17 |
| + $\mathcal{L}_g$ + $\mathcal{L}_{rd}$+ $\mathcal{L}_{sd}$+$\mathcal{L}_{pd}$ | **45.36** |

**Excluding the effect of model size.** To demonstrate that the performance improvement of the baselines are due to the effective design of the SSA-Seg rather than the introduction of additional parameters and computational consumption, we carry out the experiments shown in Table 10. Specially, *Baseline+Conv* means we add several convolutional layers to the original SeaFormer-L decoder in order to have the same model size as SeaFormer-L+SSA-Seg. Note that although SSA-Seg has more parameters, it exhibits better computational efficiency and latency, which is more important for the efficiency of the model. The experimental results show that such a significant performance improvement (*i.e.*, +3.0%) cannot be obtained by simply increasing the baseline parameters and computational consumption.

Table 10: Experiments on whether performance growth is due to model size increase.

| Method | Params | FLOPs | Latency | mIoU |
|--------|--------|-------|---------|------|
| SSA-Seg | 14.2 | 6.4 | 29.9 | 45.4 |
| Baseline+Conv | 14.0 | 6.5 | 30.4 | 42.9 |

**Effect of SSA-Seg on training.** In addition, to further validate the impact of SSA-Seg on model training, we use SeaFormer-L as the baseline model to evaluate the mIoU boost of SSA-Seg on the

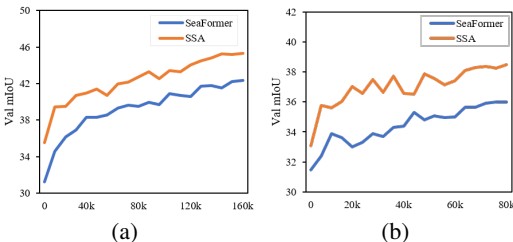

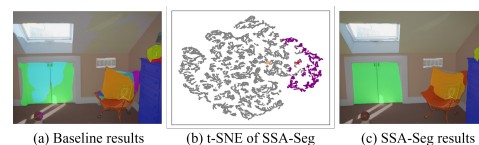

(a) Baseline results    (b) t-SNE of SSA-Seg    (c) SSA-Seg results

Figure 4: mIoU of the validation set on (a) ADE20K and (b) COCO-Stuff-10K with iterations.

Figure 5: Comparison of SSA-Seg and Baseline (SeaFormer-L) results. Purple and gray indicate pixel features in the door and other categories, respectively. Orange star indicates the initial fixed prototype of wall category, and red star indicates the adapted semantic prototype.

validation set with different iterations, as shown in Fig. 4. It can be observed that compared to the baseline model, our mIoU increases more significantly and only requires 1/3 of the iterations to achieve the same segmentation performance as the baseline model. This indicates that the SSA-Seg accelerates model learning and improves model performance. Note that due to the introduction of the teacher classifier, we need an average of 1.1 times the training time compared to the original model. However, the teacher classifier is removed when deploying the model, thus does not affect the efficiency.

**Visual comparison of SSA-Seg and Baseline results.** We provide the visualization results of SSA-Seg, which serves as a complementary illustration to Fig. 1, to further demonstrate that the SSA-Seg facilitates the adaptation of the fixed prototypes. As shown in Fig. 5, SSA-Seg effectively facilitates the offset of the prototype towards the center of the semantic domain (Fig. (b)). Since the adapted prototype is closer to the pixel features of the corresponding category, model can perform classification decisions more efficiently. As a result, SSA-Seg obtains more accurate segmentation masks compared to the baseline (Fig. (c) vs. Fig. (a)).

**Examples of extreme offsets without distillation.** We provide a visual sample to demonstrate our claim in 3.2.1 *i.e.*, the offset will even be in the direction away from the semantic features. As shown in Fig. 6, purple and gray indicate pixel features in the wall and other categories, respectively. Orange star indicates the initial fixed prototype of wall category, and red star indicates the adapted semantic prototype. It can be observed that without distillation, the offset of the prototype is uncontrollable and in extreme cases moves away from the semantic features of the corresponding image, resulting in more pixels belonging to the wall category being misclassified as curtains. Therefore, we introduce online multi-domain distillation to constrain the adaptation of the semantic and spatial prototype.

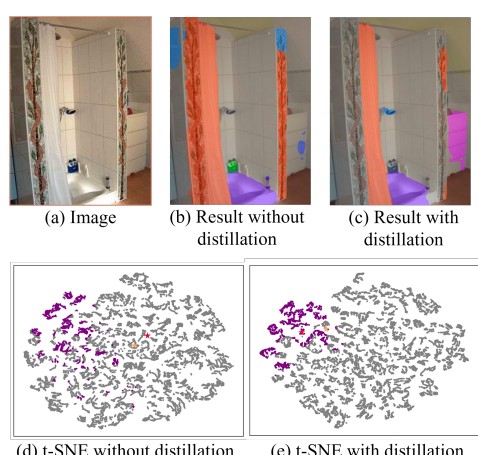

(a) Image    (b) Result without distillation    (c) Result with distillation

(d) t-SNE without distillation    (e) t-SNE with distillation

Figure 6: Examples of extreme offsets without distillation.

## 5 Conclusion

In this paper, we analyze that current pixel-level classifiers for semantic segmentation suffers limitations such as feature deviation in the semantic domain and information loss in the spatial domain. To this end, we propose a novel Semantic and Spatial Adaptive Classifier (SSA-Seg). Specifically, we employ the coarse masks obtained from the fixed prototypes as a guide to adjust the fixed prototype towards the center of the semantic and spatial domains in the test image. In addition, we propose an online multi-domain distillation learning strategy to guide the adaption process. Experimental results on three publicly available benchmarks show that the proposed SSA-Seg significantly improves the segmentation performance of the baseline models with only a minimal increase in computational cost. In particular, SSA-Seg boosts the lightweight model to achieve state-of-the-art performance. The superior performance proves that SSA-Seg can replace the vanilla pixel-level classifiers and thus contribute to the semantic segmentation research.

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

In the appendix, we provide the following items that shed deeper insight on our contributions:

- §A: Dataset and Implementation Details.
- §B: More experimental results.
- §C: More qualitative visualization.
- §D: Extra analysis

# A  Dataset and Implementation Details.

## A.1  Datasets

**ADE20K** [61] is a challenging segmentation dataset, which contains about 20,000 images and covers 150 categories. All images are annotated with pixel-level objects and object parts labels. The training set, validation set, and the test set contain 20210, 2000, and 3352 images respectively.

**PASCAL-Context** [39] is a common semantic segmentation dataset, which contains 10100 images. The train dataset contains 4996 images and test set contains 5104 images. 59 categories are labeled in this dataset.

**COCO-Stuff-10K** [1] is an extension of the coco dataset, which labels pixel-level objects. It contains 172 categories with 80 things, 91 stuff and 1 unlabeled class. There are 9K/1K images for training and testing, respectively.

## A.2  Implementation details

We use MMSegmentation [12] and follow the common training settings. During training, we apply data enhancement sequentially by random horizontal flipping, random resizing with a scale between 0.5 and 2.0, and random cropping. For ADE20K and COCO-Stuff-10K, we have a cropping size of $512 \times 512$, while for PASCAL-Context, we have a cropping size of $480 \times 480$. In addition, the batch size of all datasets is 16, and the total iterations for ADE20K, COCO-Stuff-10K and PASCAL-Context number are 160k, 80k and 80k, respectively. When inference is performed, we use whole-image inference mode. We use the mean intersection and merger ratio (mIoU) as main metric to evaluate the performance. For the chosen baseline model, we fully follow the experimental configurations such as learning rate and optimizer of the original model. During testing, we use single-scale (SS) test strategies for fair comparison. In the implementation of this paper, $\lambda_r$, $\lambda_p$, $\lambda_s$ are all set to 1, and the edge size of boundary is set to 4. Note that for SSA-Seg-related experiments, we choose different random seed replications for three times and select the corresponding intermediate values as the final evaluation metric values. The experimental error is less than 0.4% .

# B  More experimental results.

**Ablation of edge size of boundary.** The size of the edges of the boundary masks affects the performance of the response domain distillation. We explore this as shown in Table 11. The results show that the optimal segmentation results are achieved when the size of the edges is set to 4. In particular, there is a slight degradation in the performance of the model when either the edge size of the boundary is increased (*i.e.*, edge size is 5) or decreased (*i.e.*, edge size is 3). We argue that a plausible explanation is that small edge size of the boundary is not sufficient to retain enough boundary information, while large edge size of the boundary does not decouple the knowledge transfer between boundary and non-boundary regions, which results in the pixel feature information of boundary regions being interfered with by non-boundary regions.

Table 11: Ablation experiment of the edge size of boundary on the ADE20K dataset.

| Edge size | mIoU |
|-----------|-------|
| 3 | 44.74 |
| 5 | 44.98 |
| 4 | 45.36 |

**Ablation of different loss combinations.** We conduct experiments on the ade20k dataset to explore the segmentation performance of different loss combinations, as shown in Table 13. We observe that only when both $\lambda_r$ and $\lambda_p$ are 1, the corresponding segmentation

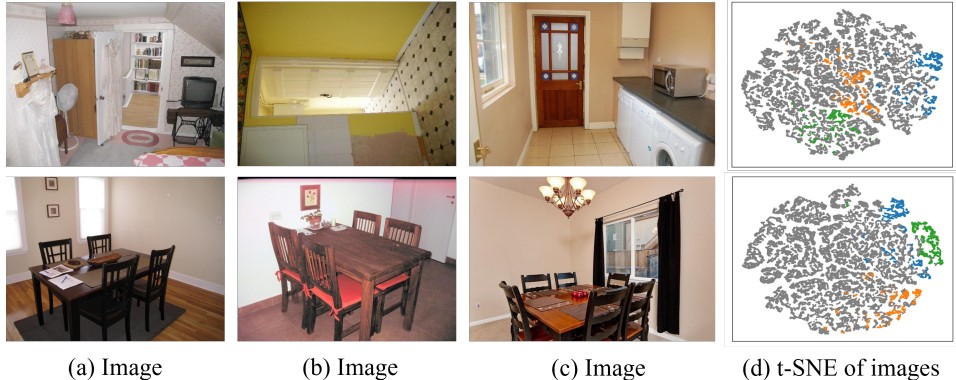

| (a) Image | (b) Image | (c) Image | (d) t-SNE of images |

Figure 7: t-SNE of some example images, which are randomly selected from the ADE20K dataset. The first row represents the distribution of pixel features in the door class, and the second row represents table class. it can be observed that due to the complex scenarios and varying object distributions, pixel features of the same class tend to exhibit larger intra-class variance when the trained model on the training set is applied to the test set.

results are relatively high, i.e., 44.67%, 44.82%, and 44.86%. Furthermore, we achieve the most superior segmentation performance when $\lambda_s$ is also 1.

**Ablation of the distillation method.** An alternative distillation strategy is to self-distill instead of using ground-truth. *self-distillation* means we use the output of the main classifier $\mathcal{O}$ to distill the coarse mask $\mathcal{M}_c$ instead of the truth mask $\mathcal{M}_g$. The results in Table 12 show that the self-distillation scheme performs poorly. We argue that the ground-truth can provide more accurate semantic information, while the segmentation results of output $\mathcal{O}$ often have some misclassified pixels, which can mislead the model's learning process from the teacher classifiers. Therefore, the prototype will offset inaccurately and impairs the segmentation performance.

Table 12: Ablation of the distillation method.

| Method | Params | FLOPs | Latency | mIoU |
|---|---|---|---|---|
| SSA-Seg | 14.2 | 6.4 | 29.9 | 45.4 |
| self-distillation | 14.2 | 6.4 | 29.9 | 43.5 |

## C More qualitative visualization.

**Visual analysis of segmentation output.** We use SeaFormer-L as the baseline to visualize the segmentation mask, as shown in Fig. 8. SeaFormer and CAC misclassify the door as the wall, and the sofas as the armchair in the second and fourth row of images, respectively. Our method has high accuracy for these confusing classes, which verifies that semantic prototype adaptation can mitigate the intra-class variance problem for correct recognition. In addition, the segmentation masks of SeaFormer and CAC for the curtain show fragmentation in the first row of images. While in the third row of images, the masks for the sidewalk show the same situation. In contrast, the shape of our segmentation masks is close to ground truth, especially in the boundary region. This validates the effectiveness of our introduction of spatial prototype adaption to model the structure of semantic objects.

**Visual analysis of CAM.** In addition, for the features output from the backbone, from top to bottom, we carry out CAM for the class of curtain, door, sidewalk, and sofa, respectively. As shown in Fig. 8, it can be seen that compared to SeaFormer and CAC, SSA-Seg is able to present stronger activation values in the center region of the mask and does not show too much activation in irrelevant regions. These qualitative analyses reflect the effectiveness of SSA-Seg in performing classification decisions.

Table 13: Ablation experiment of different loss combinations on the ADE20K dataset. The baseline is SeaFormer-L.

| $\lambda_r$ | $\lambda_s$ | $\lambda_p$ | mIoU |
|---|---|---|---|
| 0.5 | 1.0 | 1.0 | 43.91 |
| 2.0 | 1.0 | 1.0 | 43.14 |
| 4.0 | 1.0 | 1.0 | 43.62 |
| 1.0 | 0.5 | 1.0 | 44.67 |
| 1.0 | 2.0 | 1.0 | 44.82 |
| 1.0 | 4.0 | 1.0 | 44.86 |
| 1.0 | 1.0 | 0.5 | 44.05 |
| 1.0 | 1.0 | 2.0 | 44.48 |
| 1.0 | 1.0 | 4.0 | 43.90 |
| 2.0 | 2.0 | 2.0 | 44.03 |
| 4.0 | 4.0 | 4.0 | 43.02 |
| 1.0 | 1.0 | 1.0 | 45.36 |

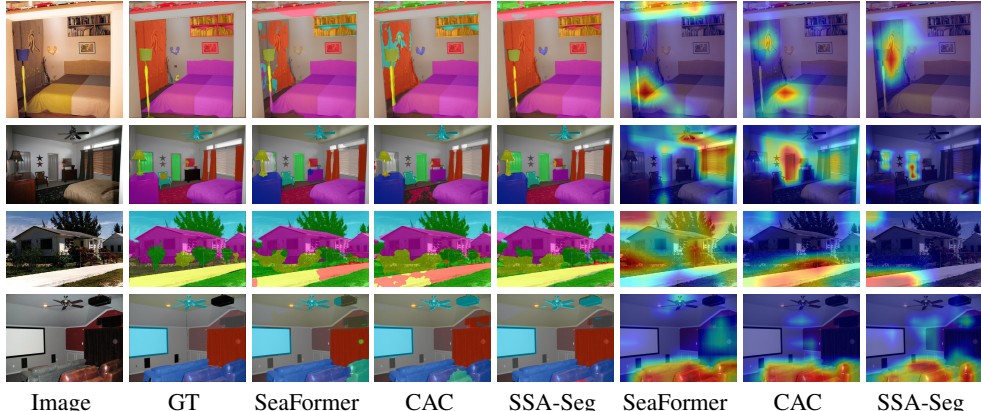

| Image | GT | SeaFormer | CAC | SSA-Seg | SeaFormer | CAC | SSA-Seg |

Figure 8: Visualization of segmentation predictions and class activation maps [41] for features output from the backbone on the ADE20K dataset. SeaFormer-L is the baseline.

**Visualization example of intra-class variance for different test images.** Compared to Fig.1, We provide additional photographic illustrations in Fig. 7 to support our statements, i.e., *Due to complex backgrounds and varying object distributions, pixel features in the test images tend to have a large intra-class variance with pixel features in the training set.*

# D  Extra analysis.

## D.1  Limitation analysis.

As a novel pixel-level classifier, SSA-Seg is able to significantly improve the performance of various baselines with a minor increase in computational cost. However, it is not compatible with mask classification methods [10, 9] due to differences in classification paradigms. Therefore, SSA-Seg cannot be applied to improve the performance of segmentation networks such as Mask2Former [9], SegViT [55], and ECENet [31]. This limits the application scope of SSA-Seg to some extent. However, when SSA-Seg is combined with efficient segmentation models such as SegNeXt [17], EfficientViT [2], it can achieve even better performance with the same or less computation compared to state-of-the-art methods. Therefore, SSA-Seg needs to rely on more advanced pixel-level semantic segmentation methods to boost its performance on general semantic segmentation tasks.

Note that in the field of mobile segmentation (or real-time segmentation), the pixel-level classification paradigm is still absolutely dominant, due to the fact that masked classification methods require multiple cross-attention to update the query, resulting in higher computational complexity and latency. Whereas SSA-Seg is able to integrate directly into existing mobile segmentation models, thus significantly improving segmentation performance with essentially no compromise on model efficiency. This validates the applicability and deployability of SSA-Seg.

## D.2  Comparison with OCRNet.

SSA-Seg is obviously different from some class-level context modeling approaches such as OCRNet [54]. First, OCRNet is a context aggregation module that enhances the representation of features based on context-aggregated class representations, but still requires a Softmax classifier (i.e., a fixed semantic prototype) to perform classification decisions. SSA-Seg, on the other hand, is a classifier whose semantic prototype is dynamic and can be offset towards the center of the semantic domain of the image. In addition, SSA-Seg can be combined with OCRNet as a classifier to enhance the performance of OCRNet, as shown in Table 1. Finally, SSA-Seg additionally introduces spatial prototype adaptation and online multi-domain distillation, which is significantly different from the work of OCRNet. Overall, SSA-Seg aims to adapt the fixed prototype towards the center of the semantic and spatial domains in the test image to solve the problem of feature deviation in the semantic domain and loss of information in the spatial domain that exists in fixed prototype classifiers.

Therefore, SSA-Seg is significantly different from existing context modeling methods [54, 56, 24], mask classifiers [9, 10], and pixel-level classifiers [28, 62, 46].

### D.3    Structural Effectiveness Analysis.

An interesting phenomenon as shown in Table 8 is that the SSA-Seg architecture (i.e., Baseline+SEPA+SPPA) achieves only marginal improvements (+0.91%), which is not an indication that the design of SSA-Seg is ineffective. Note that both the semantic prototype adaptation and spatial prototype adaptation processes involved in SSA-Seg need to be constrained by the corresponding distillation loss. Otherwise, the offset of the corresponding prototype will be uncontrollable. In extreme cases, the offset will even be in the direction away from the semantic/spatial features, as shown in Fig. 6. Therefore, the adaptation process of SSA-Seg in the semantic or spatial domain needs to be combined with the corresponding distillation loss, and the two are closely linked and inseparable. As shown in Table 8, both the adaptation of semantic prototypes (+1.9%) and the adaptation of spatial prototypes (+2.12%) significantly improve the segmentation performance of the baseline model, which validates the effectiveness of the SSA-Seg architecture design.

### D.4    Complementary analysis of SPPA.

Please note that the SPPA does not assume that objects of the same class should all be in a specific region of the image. Although SPPA proposes the concept of spatial domain centers, our spatial domain centers are obtained based on the Conditional Positional Vocoding (CPVT) [11], which represents for relative segments rather than absolute anchor points. Therefore, our spatial domain center models the collection of information about neighboring segments belonging to the same object, rather than an absolute one point. In other words, it enables the model to indirectly take into account the semantic features of pixels at neighboring locations when classifying them. Second, many images in ADE20k and COCO-stuff have the same category scattered in different areas. Nevertheless, our SSA-Seg can still significantly improve the performance of the model on these datasets. This is because SPPA takes into account the spatial structure information. We have conducted ablation experiments on SPPA in the paper, as shown in Table 8. When SPPA and distillation loss are applied, the model accuracy increased by 2.12% mIoU. This validates the effectiveness of spatial domain adaptation.

