# OpenReview forum: "SSA-Seg: Semantic and Spatial Adaptive Pixel-level Classifier for Semantic Segmentation"
_NeurIPS.cc/2024/Conference — NeurIPS 2024 poster_

### Official Review · Reviewer_Dw1G · 2024-07-12

**Soundness:** 3
**Presentation:** 3
**Contribution:** 3
**Rating:** 5
**Confidence:** 3

**Summary:**

In this paper, it analyzes that current pixel-level classifiers for semantic segmentation suffers limitations such as feature deviation in the semantic domain and information loss in the spatial domain. To this end, the authors propose a novel Semantic and Spatial Adaptive (SSA) classifier. Specifically, the authors employ the coarse masks obtained from the fixed prototypes as a guide to adjust the fixed prototype towards the center of the semantic and spatial domains in the test image. In addition, the authors propose an online multi-domain distillation learning strategy to guide the adaption process. Experimental results on three publicly available benchmarks show that the proposed SSA significantly improves the segmentation performance of the baseline models with only a minimal increase in computational cost.

**Strengths:**

The main strengths are as follows:
1. A semantic and spatial adaptive (SSA) classifier is proposed, which facilitates the offset of the fixed prototype towards the center of the semantic domain and the center of the spatial domain of the test image.
2. This paper designs multi-domain knowledge distillation to enhance the primary classifier from different domains. First, the response domain distillation distills the outputs of the two classifiers based on a boundary-aware and category-aware distillation loss, which conveys accurate semantic and structural information in the ground truth.
3. The proposed method significantly improves the segmentation performance of the baseline model  with a very small increase in computational cost on three commonly used datasets: ADE20k, PASCAL-Context, and COCO-stuff-10K.

**Weaknesses:**

The main weaknesses are as follows:
1. Why not evaluate the proposed methods on high resolution dataset, like Cityscape. How about the memory usage comparison? what's the input image resolution for training and inference? These are key information need to put in this paper.
2. For the online multi-domain Distillation, in line 199 "we first create a new branch with pixel", what does the "new branch" mean? There is no detail for the "Teacher classifier".

**Questions:**

No. please see weakness part.

**Limitations:**

No. please see weakness part.

---

> ### Author Rebuttal · Authors · 2024-08-07
>
> We sincerely thank the reviewer for their time and efforts in reviewing our work and providing valuable feedback that can further strengthen our manuscript. Below please find our detailed responses:
>
> #### **Results on high resolution dataset.**
>
> ------
>
> Due to page limits, we chose three widely used benchmark datasets for our experiments, i.e., ADE20K, COCO-Stuff-10K and PASCAL-Context. Following your comments, we have conducted several experiments on the CityScape dataset. FLOPs and Latency are calculated with the input resolution of 1024x2048 . We will add more baselines in the revised version.
>
> |                 | Params (M) | FLOPs (G) | Latency (ms) | mIoU |
> | --------------- | ---------- | --------- | ------------ | ---- |
> | SegNext-T       | 4.2        | 48.5      | 143.5        | 79.8 |
> | SegNext-T+SSA   | 4.5        | 48.6      | 145.4        | 81.2 |
> | SeaFormer-B     | 8.6        | 14.1      | 33.0         | 77.7 |
> | SeaFormer-B+SSA | 8.8        | 14.2      | 34.3         | 79.5 |
>
> As shown in the table, SSA also has a significant improvement in mIoU on the Cityscape dataset. For example, when SeaFormer-B is used as the baseline, SSA possesses an mIoU boost of 1.8 and does not affect the efficiency of the model. This is due to the fact that SSA mainly performs convolutional operations on the prototype and is less affected by the image resolution. Therefore, the efficiency and performance is more prominent when applied to high-resolution images.
>
> #### **memory usage comparison**
>
> ------
>
> Memory usage metrics have received little attention on segmentation tasks. For example, recent segmentation methods including SegViT (NeurIPS2022), CGRSeg ( ECCV2024), PEM (CVPR2024), SeaFormer (ICLR2023), and EfficientViT (ICCV2023) all do not give memory usage.  Therefore, for ease of comparison with the previous method, we do not give the memory usage. In addition, to answer your question, we have done a simple test using SeaFormer-L as the baseline, and the required memory is 4479M and 4511M respectively. This further verifies the relevant content of lines 247-250 of the paper. We will subsequently count the memory usage of the model under more baselines and add them to the revised version.
>
> In addition we have also added the comparison of parameters as shown in table 4 of rebuttal. In conclusion, the extra parameters, FLOPs, Memory and Latency brought by SSA are all negligible.
>
> #### **Input image resolution for training and inference**
>
> ------
>
> The resolution of the input images and the experimental settings are described in lines 523-535 of the Appendix. Specifically, when training, for ADE20K and COCO-Stuff-10K, we have a cropping size of 512 × 512, while for PASCAL-Context, we have a cropping size of 480 × 480. When inference is performed, we use whole-image inference mode. We'll add it to the main paper later.
>
>
>
> #### **Detail of teacher classifier**
>
> ------
>
> The architecture of the teacher classifier is shown in Fig. 2 and details can be found in the caption of Fig. 2. Specially, the architecture of the teacher classifier is identical to the main classifier, with the difference that it takes the truth mask $M_g$ as input, while the main classifier takes the coarse mask $M_c$ as input. We will add a more detailed description for lines 198-202 of the paper in the revised version.

---

### Official Review · Reviewer_bEdU · 2024-07-14

**Soundness:** 2
**Presentation:** 2
**Contribution:** 2
**Rating:** 2
**Confidence:** 5

**Summary:**

This work primarily focuses on semantic segmentation. Specifically, a semantic and spatial adaptive (SSA) classifier is proposed to address  the "feature deviation in the semantic domain and information loss in the spatial domain" issues. The proposed classifier mainly consists of

- Semantic Prototype Adaptation,
- Spatial Prototype Adaptation,
- Online Multi-Domain Distillation Learning.

The authors claim that "experimental results on three publicly available benchmarks show that the proposed SSA significantly improves the segmentation performance of the baseline models with only a minimal increase in computational cost."

**Strengths:**

The strengths of this paper can be summarized as follows,

- Overall, the author clearly introduces the research motivation and the proposed method.

- The author validates the effectiveness of the proposed algorithm across different datasets and segmentation frameworks.

**Weaknesses:**

The weaknesses of the paper are as follows,

- The authors claim in the abstract that the code for the proposed algorithm is included in the supplementary materials, but I did not find any related code there.

- Some expressions in the paper feel redundant, e.g., "directly classify masks by directly learning query vectors".

- In the introduction section, the authors argue that previous mask-level classification models have cumbersome decoders. Therefore, the authors should conduct a comprehensive comparison (params, fps, flops, miou, etc) between their method and these previous methods in the experiments section.

- Tables 1 to 3 only provide the FLOPs metric but do not include FPS and parameters, which are also crucial for deploying the model in resource-constrained environments.

- Can the authors provide any experiments or references to support the statement "Due to complex backgrounds and varying object distributions, pixel features in the test images tend to have a large intra-class variance with pixel features in the training set"? I think that a single image in Figure 1 can not prove this point.

- To be honest, Figure 1 is very confusing to me. For example, what do the gray dots represent? (pixel features belonging to other categories?) Can you provide the model's prediction results for this image? If your visualization is accurate, the prediction should include many pixels of other categories being classified as “door”. In other words, I believe there is a high likelihood that your visualization is unreliable, namely, in the t-SNE visualization results, being close to a semantic prototype does not necessarily mean that the model classifies it as the corresponding semantic category in the feature space.

- If the proposed SSA strategy truly facilitates “the offset of the fixed prototype towards the center of the semantic domain and the center of the spatial domain of the test image,” why don't the authors directly visualize this result using t-SNE like Fig. 1(b) instead of presenting a less credible schematic diagram?

- I disagree with the content in lines 36 to 40 of the paper. Modeling the relationship between prototypes and pixel features in the spatial domain is incorrect. This means that when classifying pixels, models need to consider which types of objects generally appear at that pixel location. This is obviously an overfitting to the current dataset and has no practical application value.

- The authors argue that "pixel features in the test images tend to have a large intra-class variance with pixel features in the training set" and their proposed method can address this issue. If this is indeed the case, the author should apply their method to cross-domain segmentation tasks to better demonstrate its effectiveness.

- The content in the Related Works section is too brief.

- In line 126, the authors mention that "Based on the above analysis", where is the analysis?

- Can the authors provide any experiments or references to support the statement between line 150-151?

- The content stated in lines 155 to 157 does not correspond with the formula.

- Section 3.2.1 essentially just integrates features similar to class prototypes and does not achieve the author's claimed “facilitating the offset of the fixed prototype towards the center of the semantic domain.”

- “utilizing the rich spatial information of the image” should be the responsibility of context modules (such as aspp and ppm) in the decoder head. The premise of section 3.2.2 is fundamentally flawed.

- Would the model's performance improve if only “randomly initialized position basis” is used? Can the authors provide related ablation experiments?

- If I understand correctly, section 3.2.2 essentially assumes that objects of the same class should all be in a specific region of the image. However, this assumption is unreasonable because, in semantic segmentation, different instances of the same class can be far apart.

- To be honest, I believe the methods in sections 3.2.1 and 3.2.2 are not highly related to the problem analyzed by the authors or the expected solutions to that problem.

- Would the model's performance improve if section 3.2.3 was replaced with self-distillation?

- Due to the existence of $\mathcal{M}_c$, the problem mentioned by the authors in the introduction section still persists in the proposed algorithm. In my opinion, the performance improvement is mainly due to the addition of more parameters and the introduction of more loss functions. While these changes have brought about performance gains, they do not align with the authors' motivation.

Due to these weaknesses, I believe this paper should be rejected.

**Questions:**

The main issues with this work have already been listed in the weaknesses box. The presence of numerous unreliable contents and my belief that the premise of the proposed algorithm is unreasonable are the main reasons I am inclined to reject this paper.

**Limitations:**

The author does not discuss the limitations of their algorithm in the paper. I have provided several suggestions for improvement in the weaknesses box, which I hope will help the authors enhance the quality of this work.

---

> ### Author Rebuttal · Authors · 2024-08-07
>
> **Code of SSA**. We download the supplementary material and confirm that the code is included.
>
> **Redundant expressions.** We will streamline the expression in the revised version with your comments.
>
> **Comparison with mask-level classification models** and **parameters comparison**. please refer to ***global rebuttal***.
>
> **Experiments or references to support the statement.** We provide more graphic illustrations in Fig.1 of rebuttal. In addation, a simple hypothesis can help you to solve this confusion. If pixel features of the same class do not exhibit large intra-class variance in the training and test sets, then the model should achieve very high accuracy on the test set. However, experimental results show that the baseline only has an mIoU of 42.4, and the IoU of door category is only 46.9. SSA can somewhat target this problem to achieve improved segmentation performance (mIou 45.4, IoU 49.7).
>
> **Explanation of the Fig.1.** Your understanding is accurate.  We provide a prediction result of the baseline as shown in Fig.2 (a) of rebuttal,which shows that many pixels belonging to door are misclassified. This is consistent with the visualization results of t-SNE. Your statement that the prediction should include many pixels of other categories classified as "doors" is not accurate. This is because they may be closer to the prototypes of other categories than to the prototypes of doors. In particular, due to the lack of constraints, the prototypes of some categories are more similar, as shown in Fig.3 of the paper.
>
> **t-SNE of the result.** The schematic diagram facilitates us to describe the technical route of the whole approach, i.e., semantic prototype adaptation (Fig. 1(c)) and spatial prototype adaptation (Fig. 1(d)).We also provide t-SNE and prediction for SSA in Fig.2 of rebuttal, which show that SSA indeed facilitates the offset of the fixed prototype towards the semantic domain center and the spatial domain center of the test image.
>
> **Overfitting to the current dataset.** I'm in disagreement with you. As far as I know, with the exception of multimodal segmentation work (e.g., SAM), current semantic segmentation methods (such as PEM (CVPR2024), and CGRSeg (ECCV2024)) are generally trained and tested on a single dataset. They also essentially fit the current dataset. Note that we significantly improve the performance of various state-of-the-art baseline models on all three commonly used semantic segmentation benchmarks, which validate the practical application value.
>
> **Apply method to cross-domain segmentation tasks.** It is not the focus of this article, but we will apply it to cross-domain segmentation tasks in future work.
>
> **Related works is brief.** We will add more descriptions of related work in the revised version.
>
> **where is the analysis.** The analysis is in Sec 3.1, i.e., the inadequacy of the vanilla pixel-level classifier (feature deviation in the semantic domain and information loss in the spatial domain) .
>
> **Experiments to support line 150-151.** We provide an example as shown in Figure 3 of rebuttal.
>
> **Content does not match the formula.** We have verified that there is no mismatch.
>
> **Explanation of the prototype offset.** I think you're misunderstanding the offset. In your terms, Sec 3.2.1 essentially just integrates features similar to class prototypes. however, the weights of the original fixed prototype S do change through this integration of features, and there is no problem with interpreting this as an offset of the prototype. In particular, we constrain this offset process through semantic domain distillation, which ensures that the prototypes are shifted towards the center of the semantic domain.
>
> **Premise of section 3.2.2.** Contextual modules such as ppm and aspp utilize spatial information to enhance the feature representation of pixels through contextual modeling. And SSA models the spatial information of semantic objects through spatial prototype adaptation and in turn influences the similarity between pixels and prototypes to optimize the classification decision process. They are different technical routes and can gain from each other, as shown in Table 1 of the paper.
>
> **Ablation of position basis.** Please refer to Table 2 of rebuttal.
>
> **Different instances can be far apart.** We have considered this before, but three aspects ensure that SSA proposes the design of SPPA.1, in the semantic segmentation dataset, only one instance for a category occurs in most of the pictures (especially the less-sample categories, such as door and sidewalk in Fig. 5 of the paper).SSA is better able to bring in the structured information of these objects.2, considering the occurrence of the same category in the pictures of different instances, we implement SPPA based on CPVT. it models spatial prototypes as relative segments rather than absolute anchors. The detailed analysis can be found in lines 314-321 of the paper.3, Ablation experiments verify the effectiveness of SPPA.
>
> **Relation of the methods and problem.** I'm confused about this. The problem we are posing is that vanilla pixel-level classifiers suffer from feature deviation in the semantic domain and information loss in the spatial domain. And we mitigated that problem with semantic prototype adaptation and spatial prototype adaptation. We believe that the relation between the problem and the method is highly relevant.
>
> **Performance with self-distillation.** Please refer to Table 3 of rebuttal. We argue that the prediction mask is not accurate, and using it as an input to the teacher classifier will introduce incorrect category features and noise.
>
> **performance gains not align with motivation.** Please refer to Table 5 of rebuttal, which shows that simply increasing the size of the model does not result in such a large performance gain. As for the loss function, it was proposed to serve the structure of the model, which is an inseparable part of SSA and one of the contributions of this paper.

---

### Official Review · Reviewer_ra2D · 2024-07-15

**Soundness:** 3
**Presentation:** 2
**Contribution:** 3
**Rating:** 5
**Confidence:** 3

**Summary:**

This paper proposes an adaptive method to improve the semantic segmentation quality. The main idea is to adaptively update the prototypes by using the coarse segmentation masks predicted by the baseline method. Both semantic and spatial prototypes are employed to achieve complementary improvement. Extensive experimental evaluation has been conducted on three public benchmark datasets,  showing consistent improvement over different baseline methods.

**Strengths:**

1. The method seems to be novel. The computation overhead of the method is negligible.

2. The results show that the proposed method can consistently improve many segmentation methods, in particular, the methods with lower latencies.

3. Extensive ablation studies are included to show the contributions of different components.

**Weaknesses:**

1. Not every loss term in the Equation 3 is ablated, for example L_c and L_dice

2. Since mask-level classifiers become more and more popular, how could be proposed method be adapted to mask-level methods such as mask2former?

**Questions:**

See weakness

**Limitations:**

Limitations are discussed in the appendix.

---

> ### Author Rebuttal · Authors · 2024-08-07
>
> We sincerely thank the reviewer for their time and efforts in reviewing our work and providing valuable feedback that can further strengthen our manuscript. Below please find our detailed responses:
>
> #### **Ablation of loss**
>
> ------
>
> We retain by default the cross-entropy losses $L_{ce}$ and $L_{dice}$, the former is an basic loss function for semantic segmentation and the latter commonly used in mask classification models for mask generation. We focus on ablation analysis of online multi-domain distillation loss in the paper, which is one of the important contributions of the paper. We have also done the ablation as shown below:
>
> | Method        | mIoU  |
> | ------------- | ----- |
> | SSA           | 45.36 |
> | -$L_{dice}^c$ | 44.91 |
> | -$L_{dice}^g$ | 45.06 |
>
> we observe that when removing $L_{dice}^c$ or $L_{dice}^g$ (the cross-entropy loss cannot be removed), the mIoU slightly decreases, i.e., from 45.36 to 44.91 and 45.06 respectively.
>
> #### **Adapt to mask-level methods**
>
> ------
>
> As pixel-level classifiers, SSA and mask classification methods belong to two different technical routes. However, SSA is lightweight enough, thus has the potential to be combined with mask classification methods. For example, SSA can be embedded behind a mask classifier to further refine the pre-classification results obtained by Mask2Former. We will explore better adaptation method in our future work to further improve the range of SSA applications.

---

> > ### Comment · Reviewer_ra2D · 2024-08-13
> >
> > Thanks for the clarifications which addressed my concerns. However, I am not very confident about my initial rating because I am not in the same field. After reading other reviews, I decided to decrease my rating to be a Borderline accept.
> >
> > In particular, I agree with reviewer bEdU that the writing of this paper could be significantly improved, especially clarifying the motivation and providing evidences of large intra-class variance with pixel features. Although the rebuttal addressed many questions of reviewer bEdU, some weakness points are still valid and make the paper less convincing. For example, reviewer bEdU points out that "this paper assumes that objects of the same class should all be in a specific region of the image", which is a strong assumption and does not always hold in real-world applications. The author response to this question is not convincing.

---

> ### Author Response · Authors · 2024-08-14
>
> ## **Sec3.2.2(SPPA) assumes that objects of the same class should all be in a specific region of the image**
>
> ------
>
> Thank you for your response. We have responded to reviewer bEdU in response to your question. However, due to the character limit, it is possible that our answer was not as convincing as it could have been. Here we provide a more detailed answer.
>
> - First, we do not assumes that objects of the same class should all be in a specific region of the image in our paper. This comment is unfounded and reviewer bEdU may not have fully understood our method. Although SPPA proposes the concept of spatial domain centers, our spatial domain centers are obtained based on the Conditional Positional Vocoding (CPVT) [1], which represents for relative segments rather than absolute anchor points. Therefore, our spatial domain center models the collection of information about neighboring segments belonging to the same object, rather than an absolute one point. We have a corresponding analysis in lines 314-321 of the paper. In other words, it enables the model to indirectly take into account the semantic features of pixels at neighboring locations when classifying them. More specifically, you can refer to CPVT [1], and we believe you can understand how we avoid modeling absolute spatial anchor points.
>
> - Second, many images in ADE20k and COCO-stuff have the same category scattered in different areas. Nevertheless, our classifier SSA can still significantly improve the performance of the model on these datasets. This is because SPPA takes into account the spatial structure information. We have conducted ablation experiments on SPPA in the paper, as shown in Table 3 of the paper. When SPPA and distillation loss are applied, the model accuracy increased by 2.12% mIoU. This validates the effectiveness of spatial domain adaptation.
>
> We hope you and AC are not misled by this erroneous comment.
>
> ## **Clarifying the motivation**
>
> ------
>
> As described in lines 3-8, 29-40 of the paper, our motivation is clear, i.e., the vanilla softmax classifier uses the inner product of pixel features and fixed prototypes to generate segmentation masks, leading to feature deviation in the semantic domain and information loss in the spatial domain problem. Therefore, we introduce semantic prototype adaptation and spatial prototype adaptation, adjust the fixed prototype to the center of the semantic and spatial domains in the test image, and then consider both semantic and spatial domains of the adaptive prototypes to complete the classification decision, which effectively mitigates the above problems.
>
> ## **Providing evidences of large intra-class variance with pixel features**
>
> ------
>
>  We provide more graphic illustrations in Fig.1 of rebuttal. Specially, Fig.1 shows that the t-SNE of some example images, which are randomly selected from the ADE20K dataset.  The first row represents the distribution of pixel features in the door class, and the second row  represents table class.  it can be observed that due to the complex scenarios and varying object distributions, pixel features of  the same class tend to exhibit larger intra-class variance when the  trained model on the training set is applied to the test set.We will add these graphic illustrations and a more detailed analysis to the revised version of the paper.
>
>
> ##  **SSA+Mask2Former**
>
> ------
>
> We have further validated your confusion. Specifically, after we embedded SSA into Mask2Former's mask classifier, the mIoU of the model increased from 47.2 to 48.5. This shows that SSA is able to adapt to existing mask classification methods.
>
> ##  **Conclusion**
>
> ------
>
> We have responded to each of the reviewer bEdU's comments. If you have additional questions or find one of our responses unconvincing, please do not hesitate to point it out. We look forward to further discussions with you.
>
> [1] Conditional Positional Encodings for Vision Transformers (ICLR 2023)

---

### Author Rebuttal · Authors · 2024-08-07

We are grateful to the reviewers for their thoughtful and constructive feedback. We are pleased that they recognized the novelty of the methodology, the completeness of the experiments and the excellent segmentation performance. In addition, each reviewer individually made some very valuable comments, which are conducive to the quality of the manuscript. Therefore, in addition to this global response, we have responded to each reviewer's comments separately and made the following changes to our manuscript based on the reviewers' suggestions.

- additional quantitative analyses, including comparisons of the number of parameters under various baselines and memory usage, and comparison with state-of-the-art mask classification methods.

- Additional ablation experiments, including dice loss, positional basis, optional self-distillation, and the exclusion of the hypothesis that the performance improvement is due to model size changes.

- Additional qualitative analyses, including visualization of intra-class feature distributions for more images, segmentation masks and feature visualizations for example images, and analysis of extreme offset examples.
- Other issues will be revised as per specific comments and responses.

Note that please refer to the supplementary pdf for some of the visualization and quantitative experimental results.


## Complementary Comparison Experiments

------

**Comparison with mask-level classification models.** We add some experiments with mask classification methods as shown in Table 1 of rebuttal. It should be noted that since the DPG module of CGRSeg is slightly conflicting with SSA, we remove it before adding SSA. Therefore, there is a slight decrease in the FLOPs and Latency of the model compared to the baseline. It can be observed that SSA improves CGRSeg-B and CGRSeg-L by 1.6 and 0.7, respectively. In particular, CGRSeg-L+SSA significantly outperforms recent mask classification methods such as YOSO and PEM at the same model size, with an improvement of 4.3 and 3.5, respectively. These experiment results validate that SSA exhibits a better balance of model efficiency and performance compared to masked classification methods on semantic segmentation tasks.

**FPS and parameters comparison.** As the main comparison experiment, we have provided the comparison of FLOPs, Latency (i.e., 1/fps) and mIoU in Table 1 of paper, which demonstrate the satisfactory performance and efficiency of SSA. In addition, we have further provide the comparison of parameters in Table 4 of rebuttal. There is a slight increase in the parameters of the model due to the introduction of a position basis and several 1x1 convolutional layers. In conclusion, the increase in parameters, FLOPs and latency due to SSA is negligible.

---

### Author Response · Authors · 2024-08-13
**Looking forward to further discussions**

Dear Reviewers,

Thank you for your effort in reviewing our paper.

We kindly remind you that the end of the discussion stage is approaching. During the rebuttal period, we made every effort to address your concerns faithfully, including more comparison experiments, more ablation studies, more visual analysis, etc. In particular, we have provided detailed answers to all the questions you have asked, and the codes in the supplement can be used to verify the validity of the SSA.

We believe we have effectively addressed most of your concerns and we look forward to your response or any further discussions.

Sincerely,

Paper3694 Authors

---

### Author Response · Authors · 2024-08-14
**Thanks for the review**

Dear Reviewer ra2D, bEdU and Dw1G,

Thank you for your insightful comments and constructive suggestions during the review process. Your expertise and attention to detail greatly improve the quality of our manuscript. We are confident that our final submission will address all of the issues you raised and reflect the improvements discussed. Once again, we sincerely thank you for your time and effort and for providing us with the opportunity to improve our work.

Best regards,

Paper3694 Authors

---

### Decision · Program_Chairs · 2024-09-25

**Decision:**

Accept (poster)

**Comment:**

This paper received split recommendations after the post-rebuttal discussion. The rebuttal adequately addressed most reviewer concerns about technical correctness and experimental validation. During the author-reviewer discussion, the authors provided extensive clarifications that led two reviewers to recommend acceptance. Reviewer bEdU chose not to engage in the discussion or to comment on the specific author’s responses, leaving their rejection recommendation unchanged. Overall, the paper presents a relatively novel and effective semantic segmentation technique that would interest the broad NeurIPS community. Reviewer comments and corrections, particularly those expressed by Reviewer bEdU, should be carefully included in the final version.